# Characteristics of Clinical Symptoms, Cerebral Images and Stroke Etiology in Vertebro-Basilar Artery Fenestration-Related Infarction

**DOI:** 10.3390/brainsci10040243

**Published:** 2020-04-20

**Authors:** Nobukazu Miyamoto, Yuji Ueno, Kenichiro Hira, Chikage Kijima, Sho Nakajima, Kazuo Yamashiro, Nobutaka Hattori

**Affiliations:** Department of Neurology, Juntendo University School of Medicine, 3-1-3 Hongo, Bunkyo-ku, Tokyo 113-0033, Japan; yuji-u@juntendo.ac.jp (Y.U.); ra-hi-.com@live.jp (K.H.); c-kijima@juntendo.ac.jp (C.K.); synakaji@juntendo.ac.jp (S.N.); kazuo-y@juntendo.ac.jp (K.Y.); nhattori@juntendo.ac.jp (N.H.)

**Keywords:** fenestration, cerebral infarction, dissection, Magnetic resonance imaging (MRI), basi-parallel anatomic scanning magnetic resonance imaging (BPAS), embolic stroke of undetermined source (ESUS)

## Abstract

Cerebral artery fenestration is a rare variant of the vascular architecture, but its existence is well documented. The common site of fenestration is the vertebra-basilar artery and it may be found incidentally with subarachnoid hemorrhage. However, fenestration-related cerebral infarction is rare. We analyzed the clinical characteristics, stroke etiology, and image findings of fenestration-related cerebral infarction of the vertebrobasilar artery. We reviewed our hospital records and previously published reports to find cases of fenestration-related cerebral infarction. We excluded those with unknown clinical features or radiological findings. We retrieved 4 cases of fenestration-related infarction from our hospital, in which vascular change, headache, vertigo/dizziness, and dissection in stroke etiology were detected. In eight previously reported cases of fenestration-related infarction, similar vascular changes were noted, but they were mainly diagnosed as embolic stroke of undetermined source. However, based on the criteria for dissection in this study, dissection as the stroke etiology was suspected in the previously reported cases. Many hypotheses have been proposed for the development of dissection, thrombus, and aneurysms in fenestration. Although an embryological and morphological study is needed, clinicians must consider basilar artery fenestration-related infarction as a differential diagnosis and intensive non-invasive image study is recommended.

## 1. Introduction

Artery fenestration begins with a common origin and splits into two distinct endothelium-lined channels, which then rejoin distally. Cerebral artery fenestration is a rare but well-known vascular variation [1]. The fenestration size/length varies from 1 mm to several centimeters [2]. Fenestration in the cerebrovascular system was first detected in the vertebral artery during autopsy in 1866, and many cases have been reported since [1]. In post-mortem studies, basilar artery fenestration is present in 5.26% [3].

Conventional non-invasive imaging techniques, such as computed tomography angiography (CTA) and magnetic resonance angiography (MRA), are widely used for assessing the vascular structure, including fenestration, to provide information about adjacent structures and location, as well as the arterial lumen morphology. CTA image quality has significantly advanced, become more accurate and sensitive, and can incisively delineate the existence of cerebral vascular fenestration (11–12.9%) [1,4]. In addition, CTA can accurately provide valuable information regarding the shape, size, and hemodynamic flow characteristics of cerebral arterial bifurcation, which change during aneurysm development. The first report of an aneurysm site in basilar artery fenestration by CTA was in 2003 [5]. Recently, using a large 5657 CTA series or 2280 MRA series, the shape, frequency, relationship with aneurysms, and anomalous origin of the anterior inferior cerebellar artery (AICA) in basilar artery fenestration were assessed [6,7]. More recently, a new three-dimensional (3D) black blood technique, called volumetric isotropic turbo spin-echo acquisition (VISTA) sequences, was developed, having the advantages of high multiple planar reconstruction and isotropic resolution, which helps to assess the wall nature of tangled intracranial arteries more accurately [8]. Using these methods, the plaques of basilar artery fenestration were demonstrated to exist mostly in the bifurcation and proximal segments [9].

In vertebrobasilar junction aneurysms, the frequency of basilar artery fenestration was reported to be 35–70% [10,11]. Basilar artery fenestration-associated aneurysms are more frequent in the elderly [12]. Most reports have suggested that this vascular variation has no clinical symptoms in healthy people other than subarachnoid hemorrhage (SAH) [12]. In most patients with basilar artery fenestration, acute SAH is the initial presentation of an aneurysm at the vertebrobasilar junction [10]. There were only two cases in which fenestration was the only malformation that coexisted with SAH [6,13]. Thus, it is unclear whether fenestration itself causes the formation of intracranial aneurysms [14].

There are few case reports of fenestration-related cerebral infarction in the posterior circulation [15,16,17]. Therefore, the aim of this case series study was to clarify the stroke etiology, clinical characteristics, and image findings of fenestration-related cerebral infarction of the vertebrobasilar artery.

## 2. Materials and Methods

This study was conducted in accordance with the Declaration of Helsinki. The independent ethics committee of Juntendo University Hospital approved this study. Between January 2016 and December 2019, 813 patients with cerebrovascular diseases, including ischemic and hemorrhagic stroke, were admitted to the Department of Neurology, Juntendo University Hospital. We retrospectively investigated stroke patients who had been identified as having vertebra/basilar artery fenestration by Magnetic resonance imaging (MRI)/A. We also added basi-parallel anatomic scanning magnetic resonance imaging (BPAS) to the MRI protocol. BPAS can visualize the outer contours of the vertebrobasilar artery, even in the presence of occlusion. BPAS-MRI was performed on a 3.0-T scanner in 20-mm thick coronal sections parallel to the clivus using the fast spin-echo sequence.

For all cases from our hospital records, we recorded the patient’s sex, age, National Institutes of Health Stroke Scale (NIHSS), vascular risk factors, such as hypertension (HT; systolic blood pressure (BP) of >140 mmHg or diastolic BP of >90 mmHg, or drug treatment for HT), diabetes mellitus (DM; a glycated hemoglobin (HbA1c) level of >6.5% by the National Glycohemoglobin Standardization Program (NGSP) definition or drug treatment for DM), dyslipidemia (DL; a low-density lipoprotein (LDL) cholesterol level of >140 mg/dL, a high-density lipoprotein (HDL)-cholesterol level of <40 mg/dL and a triglyceride level of >149 mg/dL or drug treatment for DL), stroke classification (TOAST classification [18]), radiological findings and the content of acute treatment/secondary prevention.

We reviewed published scientific reports by searching the PubMed database. The keywords used were “fenestration”, “cerebral infarction”, “basilar artery”, and “vertebral artery”. We reviewed related articles for all cases, and excluded “middle cerebral artery fenestration” and “double/dual vertebral artery” due to differences in symptoms, pathophysiology, and prognosis. We also excluded cases before 2000 because MRA was not common and there were mechanical differences. The same variables were collected from the previously reported cases if available (e.g., vascular risk factors, demographic data, initial treatment/final antiplatelet therapy, and MRI/MRA findings).

Fenestration classification in the basilar artery, established by Tanaka et al. [7], was used in this study, where Type I: fenestration located proximal to the AICA, Type II: bilateral AICA symmetrically originating from the fenestrated trunk, Type III: unilateral AICA originating from one side of the fenestrated trunk, and Type IV: fenestration located distal to AICA.

Cervico-cephalic arterial dissection diagnostic criteria were as follows [19]: Major criteria: (1) ‘Double lumen’ or ‘intimal flap’ demonstrated on either digital subtraction angiography (DSA), MRI, MRA, CTA, or duplex ultrasonography; (2) ‘Pearl and string sign’ or ‘string sign’ demonstrated on DSA; (3) Pathological confirmation of arterial dissection; Minor criteria: (4) ‘Pearl sign’ or ‘tapered occlusion’ demonstrated on DSA; (5) ‘Pearl and string sign’, ‘string sign’ or ‘tapered occlusion’ demonstrated on MRA; (6) ‘Hyper-intense intramural signal’ (corresponding to intramural hematoma) demonstrated on T1-weighted MRI; Additional criteria: (7) Change in arterial shape on either DSA, MRI, MRA, CTA or duplex ultrasonography; (8) No other causes of arterial abnormalities. Definite dissection was defined as the presence of one or more major criteria, or the presence of one or more minor criteria and both of the two additional criteria. Probable dissection was defined as the presence of one or more minor criteria.

## 3. Results

We extracted four cases from our hospital and eight from the literature search. The clinical presentation, radiological findings and vascular risk factors for the 12 cases are described in the Table 1.

The mean age was 46.3 ± 25.6 years (range, 5–76 years) and most of the patients were male (91.7%). Half of the patients had headache (50%). Regarding fenestration classification, type I was the most common (33.3%). In MRA study, string shape (83.3%) and shape change (91.6%) were the most common. For vascular risk factors, hypertension was the most common (50%), but no patient had diabetes (0%). For clinical symptoms, vertigo/dizziness were the most common (83.3%), and 33.3% patients had ataxia, limb weakness, and dysesthesia. Symptoms worsened in only one patient (8.3%).

In our cases, radiological follow-up was performed and vascular changes were detected in all cases. As for the stroke etiology, cerebral dissection was mainly diagnosed; however, in the reported cases, vascular change was observed in seven out of eight cases, and the stroke etiology was arterial mechanical compression in one case, dissection in one case and embolic stroke of undetermined source (ESUS) in six cases. After adjusting for the dissection criteria, four of the six previously reported cases of ESUS may have been dissection.

Below, we present our four cases, which were clinically similar in terms of dissected site (in union site) and symptoms. All cases had string sign on MRA and no recurrence.

### 3.1. Case 1

A 71-year-old female treated for dyslipidemia and hypertension developed severe headache in the left posterior region, dysarthria, vertigo, and lower limb weakness. On arrival, she had left lower limb ataxia, but no dysarthria, nystagmus or weakness on neurological examination (NIHSS 1). Brain MRI/A demonstrated a dotted acute lesion in the pons and cerebellar hemisphere (Figure 1A), reduced left vertebral artery visualization with string sign and basilar artery fenestration (Figure 1B–D). Her blood pressure was high (193/80 mmHg) without arrythmia. Laboratory data were normal, but the FDP-D-dimer level was slightly increased (2.1 mg/mL). Cardiac/carotid echography and ECG also detected no abnormality. Based on BPAS and the presence of severe headache, we diagnosed her with cerebral infarction due to cerebral dissection, and started antiplatelet therapy and blood pressure control. After starting treatment, her symptoms and visualization of the vertebral artery improved (Figure 1E).

### 3.2. Case 2

A 66-year-old man with well-controlled hypertension felt dizziness and headache, but he stayed home. The next day, his blood pressure became high (200/100 mmHg), dizziness and headache worsened, and walking became difficult. He was brought by ambulance to our hospital due to walking difficulty. His blood pressure was high (166/78 mmHg) with a regular rhythm, but ECG and cardiac/carotid echography were normal. Neurological examination revealed left gaze-induced nystagmus and left limb ataxia (NIHSS 2). On MRI/A, acute infarction was detected at the left ventral medulla and left posterior cerebellar artery area (Figure 1F). Basilar artery fenestration (with intramural hematoma; Figure 1G) was found and the left vertebral artery tapered against the union of the basilar artery (Figure 1H). Laboratory data were normal. We diagnosed him with cerebral infarction due to left vertebral artery dissection, and started antiplatelet therapy and blood pressure control. His symptoms and left vertebral artery visualization improved on follow-up MRA (Figure 1I).

### 3.3. Case 3

A 60-year-old male with untreated hypertension and smoking history developed headache, severe dizziness, and nausea at night. He vomited 10 times and was unable to walk. He was admitted to our emergency room. On arrival, his symptoms improved, but his blood pressure was 154/92 mmHg without arrhythmia. Laboratory data, ECG, and cardiac/carotid echography were normal. Neurological examination detected slight gaze-induced nystagmus (NIHSS 0). On MRI/A study, dotted acute infarction was noted in the right cerebellar area (Figure 1J), and fenestration was found on union of the right vertebral artery and basilar artery without intramural hematoma (Figure 1K–M). He was diagnosed with acute cerebral infarction due to right vertebral artery dissection, and antiplatelet therapy and blood pressure control were started. On the following MRI/A, we observed the right vertebral artery, but the intimal flap was still detected in the right fenestrated area (Figure 1N; arrow).

### 3.4. Case 4

A 71-year-old man suddenly developed headache, nausea, and hypoesthesia in the right limbs. He had hypertension and atrial fibrillation (Af), but his adherence to medication was poor. After 3 h of rest, his symptoms worsened and he was admitted to our emergency room. On arrival, his blood pressure was 182/106 mmHg with Af, and right central-type facial palsy and sensory disturbance (8/10) in the right limbs were detected on neurological examination (NIHSS 2). Laboratory data were normal, including FDP-D dimer levels. The left atrium was slightly dilatated on echocardiography, but carotid echography was normal. On MRI/A, a high-intensity area was detected in the left dorsal and ventral medulla (Figure 1O), and the left vertebral artery tapered off against the basilar artery with intramural hematoma, but fenestration was detected on BPAS (Figure 1P–R). We started hydration and direct oral anticoagulants (DOAC) were restarted after confirming no hemorrhagic complications on CT the next day. After starting treatment, facial palsy improved, but sensory disturbance remained. On follow-up MRI/A, the fenestrated vertebrobasilar artery was clearly detected, but the left fenestrated artery still had an intimal flap (Figure 1S).

## 4. Discussion

The exact underlying mechanism of fenestrated-related infarction remains unknown, but it may be related to dissection and local embolism. In our cases, dissection was mainly diagnosed, but the cases found by literature search were mainly diagnosed as ESUS. This may have been due to racial differences. Common sites of dissection may differ between Asian (posterior circulation) and Caucasian (anterior circulation) populations [25]. However, among the previously reported cases, several were suspected to be dissection.

Many hypotheses have been proposed for the development of dissection/aneurysms at the junction of a fenestrated basilar artery. Elastin discontinuity with thinned sub-endothelium at the proximal end of the fenestration may be important for aneurysm formation, similar to cerebral artery bifurcations [26]. Based on neuropathological analysis, dissection generally disrupted the internal elastic lamina and the media, similar to aneurysms [27,28]. Thus, different patterns of intimal injury were found in intracranial artery dissection. Mural hematoma may be caused by one entrance in the pseudo lumen (so-called entry-only lesions) or an entrance and exit in the pseudo lumen (so-called entry-exit lesions). Entry-only lesions have a higher rate of subarachnoid hemorrhage than entry-exit dissection lesions [29]. In addition to the micro-histological factors, the hemodynamic effects, to which the arterial wall of the proximal portion in the fenestration is directly exposed, such as the shearing stress from the blood pressure, also play an important role in the development of dissection and aneurysms [25,30].

However, most fenestrations do not directly interact or associate with the dissection/aneurysm [6,27]. Fenestration and aneurysm are not particularly complex vascular anomalies, but their incidental association has been reported based on extensive examination of SAH cases [31]. In another report, 10 of 53 patients with fenestration had associated aneurysms, but none were located at the vertebrobasilar junction with basilar artery fenestration; a significant association was not established [4]. The reported incidence of coexisting fenestration and aneurysm varies from 27.5% [1] to no association [4]. However, Sun et al. [32] found that the frequency of intracranial aneurysms with fenestration was 17.0%, with a significant difference compared with aneurysms unrelated to fenestration. In contrast, Gao reported that no aneurysms were located on the fenestration itself. Moreover, although no significant relationship exists between basilar artery fenestration and aneurysms, basilar artery fenestration was significantly more often associated with aneurysms of the posterior cerebral circulation [6].

According to the hydromechanical theory, pressure from blood is placed on the vessel wall during flow, involving wall shear stress and wall pressure. At the blood vessel branching site, the interior wall has high wall pressure and wall shear stress, whereas the lateral wall has low wall pressure and wall shear stress, with an irregular distribution and high pressure gradient [9]. As the atherogenic components in the blood move passively from the high-pressure area to the low-pressure area, low wall pressure is not only closely related to the blood vessel wall thickness, but also to atherosclerosis [9]. When there was a more intensive pressure change, these components significantly aggregated. Thus, hydromechanical pressure affects the vascular endothelium [33].

Our study has several limitations, including its retrospective design and the non-random treatment allocation procedure. It must be emphasized that our study was exploratory in nature and is therefore appropriate for hypothesis generation, but not for hypothesis testing. Second, we were unable to retrieve information about the duration of treatment or compliance with daily medication considering the retrospective nature of this study. Furthermore, the medical records of a substantial number of patients contained no information about long-term outcomes, or the post-discharge use of anti-platelet therapy, anti-hypertensive therapy or statins. Third, selection bias may have affected our analysis. As basilar artery occlusion affects respiration and hemodynamics, MRI was unable to be performed for those patients. Therefore, fenestration was overlooked. Indeed, in a case reported by Belly et al., fenestration was detected by post-modern analysis [34]. However, we believe that this study is useful as it may lead to treatment methods for vertebra-basilar artery fenestration-related cerebral infarction.

## 5. Conclusions

In conclusion, we reported four cases of rare fenestration-related cerebral infarction. In our cases, MRI/A analysis was performed in detail and during follow-up. Focusing on stroke etiology, cerebral artery dissection was suggested, but we were unable to exclude underlying etiologies. MRI/A and BPAS may be useful for observing the fenestrated vascular structures.

## Figures and Tables

**Figure 1 brainsci-10-00243-f001:**
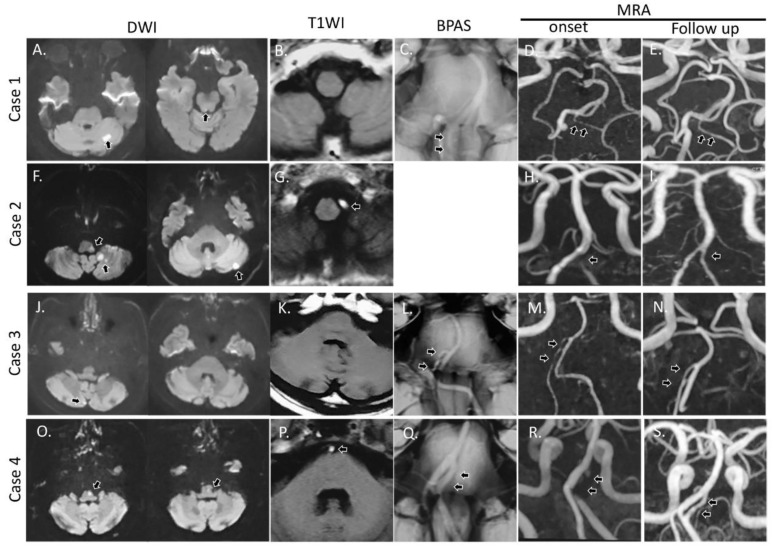
Magnetic resonance imaging (MRI) of our cases on diffusion weighted image (DWI; **A**, **F**, **J**, **O**), T1WI (for intramural hematoma; **B**, **G**, **K**, **P**), basi-parallel anatomic scanning magnetic resonance imaging (BPAS; **C**, **L**, **Q**) and magnetic resonance angiography (MRA, onset; **D**, **H**, **M**, **R**, follow-up; **E**, **I**, **N**, **S**). Case 1; Acute infarction at the left cerebellum and dorsal pons (**A**; arrow), but no intramural hematoma. The left vertebral artery was observed on BPAS (**C**; arrow), but not on MRA (**D**; arrow). On follow-up MRA, the left vertebral artery was observed (**E**; arrow). Case 2; Acute infarction was observed in the lateral medulla and cerebellum (**F**; arrow). Intramural hematoma was noted at the left vertebral artery (**G**; arrow). On initial MRA, string sign was seen in the left vertebral artery (**H**; arrow). On follow-up MRA, the left vertebral artery was visualized (**I**; arrow). Case 3; Dotted acute infarction was noted in the right cerebellum (**J**; arrow), but there was no intramural hematoma (**K**). On BPAS and MRA, the right fenestrated artery and vertebral artery were string-shaped (**L**, **M**; arrow). On follow-up MRA, the right fenestrated artery and vertebral artery were visualized more clearly. Case 4; Acute infarction was observed at the dorsal lateral medulla (**O**; arrow) and intramural hematoma was present at the left fenestrated artery (**P**; arrow). On BPAS, the basilar artery was fenestrated (**Q**; arrow), but the left fenestrated artery disappeared on initial MRA (**R**; arrow). On follow-up MRA, the left fenestrated artery reappeared (**S**; arrow).

**Table 1 brainsci-10-00243-t001:** Clinical features, vascular risk factors and radiological findings of vertebra-basilar artery-related infarction.

Author (et al.)	Age	Sex	Stroke Etiology	Fenestration Classification	Headache	Infarct Area	Findings Suggesting Dissection on MRA				Intramural Hematoma on T1WI	D-Dimer (μg/mL)	Hypertension	Dyslipidemia	Diabetes	Atrial Filiation	Related Injury	Worsening	Vertigo/Dizziness	Ataxia	Weakness	Dysesthesia
							String	Pearl	Double Lumen	Shape Change												
Case 1	71	F	D	I	+	pons, cerebellar	+	-	-	+	-	2.1	+	+	-	-	-	-	+	+	-	-
Case 2	74	M	D	III	+	Medulla PICA	+	-	-	+	+	<1.0	+	-	-	-	-	-	+	+	-	-
Case 3	61	M	D	III	-	cerebellar	+	-	-	+	-	1.1	+	-	-	-	-	-	+	-	-	-
Case 4	71	M	D	I	+	medulla	+	-	-	+	+	1.5	+	+	-	+	-	-	+	-	+	+
Bernard [20]	18	M	D	(VA)	+	cerebellar	+	+	-	+	NA	NA	-	-	-	-	-	+	+	-	-	-
Gold [16]	12	M	E	II	-	cerebellar	-	-	-	NA	NA	NA	-	-	-	-	-	-	+	+	-	-
Kloska [15]	5	M	E	I	+	pons	+	-	-	+	NA	NA	-	-	-	-	-	-	-	-	+	-
Meinnel [21]	76	M	E	IV	-	pons	+	-	-	+	NA	NA	+	+	-	-	-	-	-	-	+	+
Palazzo [17]	56	M	E	I	-	cerebellar posterior lobe	+	-	-	+	NA	NA	+	-	-	-	-	-	+	+	-	-
Wu [22]	36	M	E	IV	-	Cerebellar (vermis)	-	-	-	+	NA	NA	-	-	-	-	-	-	+	+	-	-
Yamaguchi [23]	45	M	C	(VA)	-	Pons cerebellar	+	-	-	+	NA	NA	-	-	-	-	-	-	+	-	-	-
Yamamoto [24]	30	M	E	(VA)	+	midbrain	+	-	-	+	NA	<1.0	-	-	-	-	-	-	+	-	+	+

NA; not applicable, MRA; magnetic resonance angiography, D; dissection, C; compression, E; embolic stroke of undetermined source, PICA; posterior inferior cerebellar artery.

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
