# Peer review of "Characteristics of Clinical Symptoms, Cerebral Images and Stroke Etiology in Vertebro-Basilar Artery Fenestration-Related Infarction"

_brainsci, 2020, doi:10.3390/brainsci10040243_

Round 1

Reviewer 1 Report

Comment: The authors present 4 cases of stroke and association with fenestration of the vertebro-basilar artery. In addition, the authors summarize 8 cases of the literature. The authors conclude that many hypotheses are discussed regarding the development of dissection, thrombus and aneurysms in fenestration. The authors advise that basilar artery fenestration-related infarction is a differential diagnosis in stroke diagnosis and care.

The manuscript is interesting and well comprehensible. The findings are for the general neurologist interesting and - taken all together -plausible. In addition, the strengths and the limitations are accurately discussed.

There are some minor concerns:

  • In the title “basilar” artery should be replaced by “vertebro-basilar”
  • The authors should describe more in detail the MRI-method BPAS
  • Page 5, line 175: what exactly would the authors like to express with “the lesions reappeared”?
  • Page 5, line 176: I would say the acute infarction is seen at the ventral lateral medulla and not at the dorsal one.

Author Response

We agreed with all of the helpful suggestions and comments from the reviewers, and tried to address all of the important points. The following text describes our responses to the comments made by the reviewers. And also, we resubmit for English proofreading service. Please see attached certification letter.

Reviewer #1

In the title “basilar” artery should be replaced by “vertebro-basilar”

We greatly appreciate your useful comments. As the referee suggested, we revised the title (line 2, page 1).

The authors should describe more in detail the MRI-method BPAS

Thank you for this important suggestion. We apologize that we did not provide appropriate explanations. We added the details of BPAS to the Methods section (line 71, page 2).

Page 5, line 175: what exactly would the authors like to express with “the lesions reappeared”?

We apologize for the incomplete explanation. We wanted to express that the right fenestrated artery and vertebral artery were visualized more clearly in Case 3 on follow-up MRA. We revised the text and the figure legends (line 179, page 5).

Page 5, line 176: I would say the acute infarction is seen at the ventral lateral medulla and not at the dorsal one.

We apologize for our mistake. We revised the text (line 180, page 5).

Once again, we are grateful for all of the helpful suggestions and for giving us the chance to submit our revised manuscript to Brain Science.

Reviewer 2 Report

Miyamoto et al. present a retrospective case series of 4 patients with stroke associated with arterial fenestration: “Characteristics of clinical symptoms, cerebral images and stroke etiology in basilar artery fenestration- related infarction” The case series is supplemented by reviewing  6 cases from the literature.

At the end, they find evidence for local dissection as a cause of stroke in all of their 4 cases as well as some cases from the literature as far as advanced  vessel imaging information were provided. Alternatively, they speculate on progression of atherosclerosis at the site of fenestration, and subsequent plaque rupture as a cause of stroke, but without giving evidence for that in their cases. Obviously, arterial fenestration does not protect from more common stroke aetiologies, as it is atrial fibrillation in one of their cases. Although arterial fenestration may not be directly causative but an epiphenomenon of local arterial abnormality (dysmorphic arterial wall composition as a risk for arterial dissection, or hemodynamics causing premature arterosclerosis with subsequent plaque rupture) it indicates a rare cause of stroke and challenges the assumption of cryptogenic stroke or “embolic stroke of unknown source” (ESUS).   

However, phrasing of the article is somewhat awkward and vague, and the differentiation between evidence based information as opposed to speculative parts remain elusive. So, a thorough re-writing with the help of a native speaker would help a lot.  

In detail:

  1. 12 “a rare normal variant, but its existence is well 12 documented.” >>a variant of vascular architecture
  2. 14:

 fenestration-related cerebral infarction is  rarer (than?) but contradicting evidence is given below, cf l. 36.: omit

  1. 27: “must be performed” better: suggest, recommend
  2. 55 “this vascular variation has no clinical symptoms”: who? patients , healthy people? The paragraph starts with “in vertebrobasilar aneursmys” so the existence of SAH is due to the aneurysma, right? The whole paragraph comes to no point.
  3. 117 NIHSS (last “S” means score!) 1, not 1 point, same below

ll.213 this paragraph intermingles the discussion of the “dissection hypothesis” with the hypothesis of “premature atherosclerosis” due to shear stress asf. That should be more clearly separated.

l.233 “it may lead to treatment methods for patients for whom an appropriate treatment as yet to be established.” remains obscure.

Author Response

We agreed with all of the helpful suggestions and comments from the reviewers, and tried to address all of the important points. The following text describes our responses to the comments made by the reviewers. And also, we resubmit for English proofreading service. Please see attached certification letter.

Reviewer #2

12 “a rare normal variant, but its existence is well documented.” >>a variant of vascular architecture.

We apologize for our mistake. We revised the text (line 12, page 1).

14: fenestration-related cerebral infarction is rarer (than?) but contradicting evidence is given below, cf l. 36.: omit

We apologize for the incomplete explanation. As the reviewer suggested, we revised the text in the abstract (line 15, page 1) and deleted the contradictory statement (line 35, page 1).

27: “must be performed” better: suggest, recommend

We apologize for the incomplete explanation. We revised the text (line 27, page 1).

55 “this vascular variation has no clinical symptoms”: who? patients, healthy people? The paragraph starts with “in vertebrobasilar aneurysms” so the existence of SAH is due to the aneurysm, right? The whole paragraph comes to no point.

We apologize for the incomplete/confused explanation. We revised the text (line 55, page 2).

117 NIHSS (last “S” means score!) 1, not 1 point, same below

We apologize for our mistake. We corrected the entire manuscript (lines 124, 138, 149 and 160).

213 this paragraph intermingles the discussion of the “dissection hypothesis” with the hypothesis of “premature atherosclerosis” due to shear stress asf. That should be more clearly separated.

We apologize for the incomplete/confused explanation. We deleted the hypothesis of premature atherosclerosis and changed this paragraph in the revised manuscript (lines 217-224, page 7).

233 “it may lead to treatment methods for patients for whom an appropriate treatment as yet to be established.” remains obscure.

We greatly appreciate your useful comments. According to the reviewer’s comment, we changed the explanation to ‘it may lead to treatment methods for vertebra-basilar artery fenestration-related cerebral infarction.’ (line 234, page 7).

Once again, we are grateful for all of the helpful suggestions and for giving us the chance to submit our revised manuscript to Brain Science.